# Effects of the In Ovo Injection of L-Ascorbic Acid on Broiler Hatching Performance [note 1]

**DOI:** 10.3390/ani12081020

**Published:** 2022-04-14

**Authors:** Ayoub Mousstaaid, Seyed A. Fatemi, Katie E. C. Elliott, Abdulmohsen H. Alqhtani, Edgar D. Peebles

**Affiliations:** 1Department of Poultry Science, Mississippi State University, Mississippi State, MS 39762, USA; am4768@msstate.edu (A.M.); katie.elliott@usda.gov (K.E.C.E.); aha107@msstate.edu (A.H.A.); d.peebles@msstate.edu (E.D.P.); 2Poultry Research Unit, USDA-ARS, Mississippi State, MS 39762, USA; 3Department of Animal Production, King Saud University, Riyadh 11451, Saudi Arabia

**Keywords:** broiler, hatchability, in ovo injection, L-ascorbic acid, serum L-ascorbic acid

## Abstract

**Simple Summary:**

Previous studies have shown positive effects of the use of supplementary L-ascorbic acid (L-AA) to mitigate various stressors such as heat and ammonia exposure in the broiler industry. The aim of the current study was to determine the effects of L-AA administrated by in ovo injection on various hatch variables and the embryonic serum L-AA concentrations of Ross 708 broilers. At 18 days of incubation (doi), the following four treatment groups: non-injected control, saline-injected control, and saline containing either 12 or 25 mg of L-AA were administrated. An automated multi-egg injector accurately delivered 100 μL solution volumes into the amnion. The in ovo injection of high levels of L-AA (12 and 25 mg) did not affect hatchability, but 12 mg of L-AA in saline and saline alone resulted in a reduction in embryonic mortality. Additionally, serum L-AA did not differ between the in ovo injected treatments at any time period; however, the serum L-AA concentration was numerically higher in males as compared to female hatchlings. In conclusion, the automated in ovo injection of high levels of L-AA may not be detrimental to hatchling quality but may promote embryonic livability.

**Abstract:**

Effects of the in ovo injection of various concentrations of L-ascorbic acid (L-AA) on the hatchability and retention levels of L-AA in the serum of broiler embryos were investigated. A total of 960 Ross 708 broilers hatching eggs were randomly divided into four treatment groups: non-injected control, saline-injected control, and saline containing either 12 or 25 mg of L-AA. At 18 days of incubation (doi), injected eggs received a 100 μL volume of sterile saline (0.85%) alone or containing one of the two L-AA levels. Percentage egg weight loss was also determined from 0 to 12 and 12 to 18 doi. Hatch residue analysis was conducted after candling to determine the staging of embryo mortality. At approximately 21 doi, hatchability of live embryonated eggs (HI) and hatchling body weight (BW) were determined. Blood samples were taken at 6 and 24 h after L-AA in ovo injection to determine serum L-AA concentrations. Serum L-AA concentrations, HI, and hatchling BW did not differ among all treatment groups. However, chicks in the non-injected group had a higher (*p* = 0.05) embryonic mortality at hatch in comparison to those in the 12 mg of L-AA in saline and saline alone treatment groups. These results suggest that the in ovo injection of high levels of L-AA (12 and 25 mg) does not negatively affect HI or serum concentrations of L-AA but has the potential to promote embryonic livability. Further research is needed to determine the retention time of L-AA in the other tissues of broilers, including the cornea of the eye, in response to different levels of supplemental L-AA.

## 1. Introduction

In nature, L-ascorbic acid (L-AA) can occur as the reduced form, which can be reversibly oxidized to dehydro-L-ascorbate. In its metabolism, L-AA is first converted to dehydro-L-ascorbate by several enzymatic or nonenzymatic processes and can then be reduced back to L-AA in cells [1]. The L-isomer is biologically active, whereas the D-isomer is not. The oxidized form can be further irreversibly oxidized to the inactive form (diketogulonic acid). Because of this, L-AA is very susceptible to destruction through oxidation, which is accelerated by heat and light.

Absorbed L-AA readily equilibrates with that already present in the body pool. No specific binding proteins for L-AA have been reported, and it is suggested that the vitamin is retained by binding to subcellular structures. Under normal conditions, poultry can synthesize L-AA within their body, and L-AA is absorbed in a manner similar to carbohydrates (monosaccharides). Intestinal absorption in L-AA-dependent animals appears to require a sodium-dependent active transport system [2]. In the developing embryo and newly hatched chick, L-AA has been shown to modulate the activities of antioxidant enzymes such as superoxide dismutase, glutathione peroxidase, and catalase [3,4]. This becomes particularly important due to the fact that the tissues of chicken embryos contain a high proportion of polyunsaturated fatty acids in the lipid fraction and, thus, need antioxidant defense. Therefore, an increase in the serum or tissue concentrations of L-AA may improve the hatchability as well as the posthatch performance of chickens.

Several studies have been conducted to determine the effects of dietary L-AA supplementation on the performance of broilers subjected to heat stress. The administration of a 12 g/100 L dosage of L-AA in drinking water has been shown to reduce oxidative stress in birds, especially in chronically heat-stressed birds [5]. In addition, L-AA supplementation has been shown to improve the performance as well as the immunity of broilers [6]. The partial effects of L-AA on the immune system of broilers may be related to the activity of phagocytes and the production of lymphocytes and cytokines, which have also been shown to be modulated by supplemental dietary L-AA [7].

In ovo injection has emerged as a method by which to promote the immunity as well as the growth of broiler embryos [8,9]. It is well documented that the in ovo injection of substances, including carbohydrates, fatty acids, vitamins, and hormones, can enhance embryonic and posthatch immunity and development at an early age [9]. The in ovo injection of various fat-soluble vitamins, including vitamin D_3_ and vitamin E, has been shown to affect the hatch and posthatch performance of broilers. The in ovo injection of various vitamin D_3_ sources has been shown to increase the hatch quality [10] as well as the posthatch performance [11,12,13] of broilers. In addition, the in ovo injection of water-soluble vitamins, including L-AA, has been shown to increase hatchability [14]. Additionally, a decrease in embryonic mortality and maximized hatchability were observed when broiler eggs were dipped in low levels of L-AA solutions [15], while improvement in hatch quality and hatchability has not been reported.

However, Zhang et al. [16] reported that there was no effect on broiler hatchability when L-AA in a 0.5 to 13.5 mg dosage range was manually administrated by hand in ovo-injection at 17 days of incubation (doi). Similar to this result, it was reported that when 3 to 6 mg of L-AA was in ovo-injected into individual eggs that hatchability was not affected [17]. Zhang et al. [4] also reported that the automated in ovo injection of 3 to 12 mg of L-AA at 17 doi had lasting positive effects on the posthatch growth performance of broilers, but that 36 mg of L-AA also has the potential to improve meat quality. However, in comparison to saline-injected controls, the injection of 6 or 12 mg of L-AA resulted in higher body weight (BW) gain and feed intake during the grower phase. The inconsistent results after an in ovo-injection of L-AA on various hatch variables may be related to differences in the level of injected L-AA, the site of injection (air cell vs. amnion), the timing of injection (between 15 and 18 doi), the mode of injection (manual vs. automated), or the amount of time (approximately 3 to 5 d) between the d of injection and time of hatch. Additionally, automated in ovo injection of high levels (12 to 25 mg) of L-AA at 18 doi and their subsequent effects on various hatchling variables have not been previously investigated. The automated in ovo injection of different substances such as vaccines, at 18 doi, has been widely used in the USA [8]. The in ovo-injection of L-AA has further been demonstrated to stimulate posthatch immunity of broiler in other studies [14,17,18].

Earlier studies have reported the effects of dietary L-AA on various blood variables [19,20], but its effect on serum L-AA concentrations and retention time in broiler embryos have not been reported. Additionally, an increase in serum L-AA has been associated with an improvement in layer performance and vitamin A concentration when fed a diet supplemented with 250 mg/kg of L-AA [21]. The serum L-AA in either broiler embryo or posthatch has not been previously determined to identify its relationship with possible improvement in hatchling or grown chickens when L-AA was in ovo administrated. In the present study, we hypothesized that in ovo injection of L-AA of 12 and 25 mg at 18 doi could stimulate serum L-AA in order to improve hatchling quality. The objectives of this study included the determination of serum L-AA concentrations and the hatching success of broiler embryos in response to the automated in ovo injection of supplemental L-AA (12 or 25 mg) into the amnion at 18 doi. Preliminary results of the current study have been previously published in an abstract form [22]. The results of this study will also provide more information concerning the potential effects of the in ovo injection of L-AA on broiler chick hatchability and hatchling quality at 21 doi, as well as the associated response in the serum L-AA concentrations of the broiler embryos at 18 and 19 doi.

## 2. Material and Methods

### 2.1. General

A total of 1440 broiler hatching eggs collected from 41-week-old Ross 708 broiler breeder hens were stored under commercial conditions for 72 h prior to set according to the procedure described by Zhang et al. [18]. Thirty eggs were randomly set in each of 4 treatment groups on each of the 8 replicate tray levels (960 total eggs) in a calibrated NMC2000 single-stage Incubator (Nature Form Incubator Company, Jacksonville, FL, USA). The 4 in ovo injection treatments were: (1) non-injected control, (2) sham-injection control (injection of 100 μL of saline), or injection of 100 μL of saline containing (3) 12 mg of L-AA (L-AA 12), or (4) 25 mg of L-AA (L-AA 25). The treatment groups were randomly assigned to the 8 tray levels in the incubator. The same incubator served as both a setter and hatcher unit [12,13]. The incubational conditions were in accordance with the procedures described by Fatemi et al. [12]. Temperature and relative humidity readings were recorded in 4 locations within the incubator every 15 min using HOBO ZW Series wireless data loggers (Onset Computer Corporation, Bourne, MA, USA). All eggs were candled at 12 and 18 doi to remove eggs that did not contain live embryos. Furthermore, the difference between egg weight prior to set and egg weight at candling was used in determining the mean percentage egg weight loss (PEWL) between 0 and 12, 12 and 18, and 0 and 18 doi in each treatment-replicate group. Mean PEWL was calculated according to the procedure of Peebles et al. [23], and PEWL between 12 and 18 doi was determined after the removal of eggs that did not contain live embryos.

### 2.2. Treatment Solutions Procedure and Injection

The treatment solutions were prepared according to the procedure described by Zhang et al. [16]. Briefly, L-AA (A92902, Sigma-Aldrich Inc., St. Louis, MO, USA) was used to freshly prepare the L-AA solutions prior to an injection using a 0.2 μm syringe filter (PTFE, 25 mm, Scientific Strategies, Yukon, OK, USA). Different levels (12 and 25 mg) of L-AA were suspended in 85% saline and mixed thoroughly until the L-AA was completely dissolved. The solutions were then refrigerated for 5 h until injection. Live embryonated eggs were injected into the amnion at 18 doi using a commercial automated multi-egg injector (Embrex Inovoject m, Zoetis, Parsippany, NJ, USA). At the time of injection, one of the eggs from each of the 4 treatment groups on each of the 8 incubator tray levels (32 total eggs) were injected with colloidal coomassie brilliant blue G-250 dye (Genlantis, San Diego, CA, USA) and immediately euthanized for embryo staging analysis.

### 2.3. Hatch and Serum Data Collection

At 20.92 doi (502 h of incubation (hoi)), mean hatchling BW for each replicate group of chicks in each treatment group, and hatchability as a percentage of live embryonated eggs (HI) were calculated. After the in ovo injection of L-AA, blood samples were collected from the chorioallantois vasculature of 4 randomly selected eggs within each treatment group on each of the 8 replicate tray levels at 18.25 doi (438 hoi), 18.67 doi (454 hoi) and at 20.92 doi (502 hoi) for both males and females. Serum from the 4 eggs in each treatment-replicate group was extracted and pooled according to the procedure described by Fatemi et al. [10]. Serum L-AA concentration was determined according to the manufacturer’s protocol (Chicken Vitamin C (VC) Elisa kit; MyBioSource, San Diego, CA, USA). The L-AA ELISA assay was performed as described by Fatemi et al. [24]. Current modifications to the procedure included the use of biotinylated chicken L-AA antibody, and specific L-AA monoclonal antibody precoated plates were used. Optical density (OD) at 450 nm (OD450) for L-AA was later determined with a SpectraMax M5 Microplate Reader (Molecular Devices, San Jose, CA, USA).

### 2.4. Statistical Analysis

The experimental design was a randomized complete block. Each of the 8 replicate tray levels in the incubator contained the 4 treatments and the tray level was considered the blocking factor. All treatments were randomly represented on each of the incubator tray levels. All variables were analyzed by one-way ANOVA using the GLIMMIX procedure of SAS 9.4© (SAS Institute, Cary, NC, USA, 2013). The results are shown as mean ± SEM, and differences were deemed statistically significant at *p* ≤ 0.05. Means separations were performed by Fisher’s protected least significant difference [25]. Analysis of the incubation data followed the model: Yij = μ + Bi + Tj + Eij, where μ was the population mean; Bi was the block factor (i = 1 or 8); Tj was the effect of each in ovo injection treatments (j = 1 to 4); Eij was the residual error.

## 3. Results

In the dye-injected eggs across all treatment groups, 91.67% and 8.33% of the injections occurred in the amnion and embryo, respectively. More specifically, in the non-injected control group, all (100%) of the dye-injected eggs received injections in the amnion, and in the dye-injected eggs across the injection treatment groups, 88.9% and 11.1% of the injections were in the amnion and body proper, respectively. Wakenell et al. [26] (2002) reported that optimal (90% protective index) efficacy of the Marek’s disease vaccine was found when injections were via amniotic or intraembryonic routes. The hatch variables, including 0 to 12, 12 to 18, and 0 to 18 doi PEWL; HI; and hatchling BW, are presented in Table 1. No significant (*p* > 0.05) treatment differences were observed for any of these variables. The hatch residue results at 21 doi are provided in Table 2. Late, pip, post-pip, and hatchling mortalities were defined, respectively, as those mortalities that occurred between 18 and 21 doi prior to pip, during the pipping process, after the pipping process, and immediately after complete emergence from the shell. Hatch residue analysis revealed that late, pip, and post-pip mortalities did not differ significantly (*p* > 0.05) among treatments. However, there was a significant treatment difference (*p* = 0.05) for hatchling mortality (Table 2). Hatchling mortality was significantly lower in the saline and L-AA 12 treatments than in the non-injected control group, with the L-AA 25 treatment being intermediate. The L-AA serum concentration results in response to the different in ovo injection levels of L-AA at 18, 19, and 21 doi are shown in Table 3. The serum concentration of L-AA did not differ (*p* > 0.05) among treatments at 18 and 19 doi. At 21 doi, no differences were observed between the in ovo injection treatments for L-AA serum concentrations (*p* > 0.05) in both males and females; however, there was a numerical difference between male and female serum L-AA concentrations in hatchlings, in which the mean value of L-AA serum concentration in males was approximately two times higher than in females.

## 4. Discussion

The objectives of this study were to determine the effects of the automated in ovo injection of high levels (12 and 25 mg) of L-AA at 18 doi on the broiler hatch variables as well as their serum L-AA concentrations at 18 and 19 doi. The results of the current study revealed that the L-AA 12 and L-AA 25 treatments did not exhibit negative effects on the HI and hatchling BW of the broilers. Similar to the current study, the in ovo injection of saline containing 0.5, 1.5, 4.5, or 13.5 mg of L-AA has been previously observed to not affect the hatchability or hatchling BW of broilers [16,17]. This indicates that a higher in ovo injection dosage of L-AA (25 mg) is safe. Dissimilar to the current findings, it has been reported that the in ovo injection of 3 mg of L-AA at 11 and 15 doi increased hatchability [18,27,28]. Furthermore, an increase in hatchability in response to the in ovo injection of L-AA was observed when 3 mg of L-AA was administrated at 15 doi into the air cell [27,29]. Inconsistencies between the results of the current and previous research may be linked to the time and site of injection, the mode of injection, or the level of injected L-AA. However, in the current study, 12 and 25 mg of L-AA were injected into the amnion at 18 doi. The results of earlier studies have shown that the in ovo injection of L-AA reduced embryo mortality rate when embryos received the in ovo injection of 3 mg of L-AA [29]. Conversely, the injection of 12 or 25 mg of L-AA had no significant effect on embryo mortality through the post-pip stage in this study.

In ovo injection of L-AA at 18 or 15 doi has been shown to increase antioxidant capacity [4,18,30] as well as cause a reduction in inflammatory gene activity in broilers at 42 d of age (doa). It is well documented that the in ovo injection of L-AA at various levels increases total superoxide dismutase and malondialdehyde antioxidant activity in Ross 708 broilers [14,31]. Similar results were also reported when an in ovo injection of 12 or 36 mg of L-AA was applied at 17 doi in Ross 708 broilers [4]. Antioxidant defense systems consist of enzymatic and non-enzymatic antioxidants components scavenges continuously free radical that has been generated in the animal body [32]. The enzymatic mechanism is involved in the conversion of superoxide anions to hydrogen peroxide, which is facilitated by Superoxide dismutase in a cellular antioxidant reaction. Additionally, detoxification of the hydrogen peroxide product is individually mediated by glutathione peroxidase and catalase independently. The non-enzymatic reaction is linked to Malondialdehyde, a major product of several unsaturated aldehydes and ketones generated from oxidative damage. It is well reported that Malondialdehyde is mostly used as an indicator for evaluation of the degree or level of lipid peroxidation. Nevertheless, a possible increase in antioxidant capacity and a possible decrease in inflammation, in association with a potentially improved immune response, did not appear to have a subsequent effect on late-stage mortality in the embryos that received in ovo injections of 12 or 25 mg of L-AA in this study.

Serum L-AA concentrations were found to not differ among the treatment groups at both 18 and 19 doi in the current study. However, an increase in serum L-AA concentration was observed in a different study in birds subjected to heat stress in response to the provision of 12 g of L-AA dissolved in 100 L of drinking water [5]. Additionally, the increase in serum L-AA concentration in response to the water source of L-AA could be linked to an increase in total superoxide dismutase and a decrease in malondialdehyde serum levels. These results indicate that increased serum L-AA levels can be associated with an improvement in antioxidant activity [33]. Conversely, despite an increase in antioxidant capacity, Del Barrio et al. [34] did not observe an increase in the serum L-AA concentrations of heat-stressed broilers when provided 200 to 1000 mg/kg of supplemental L-AA in their drinking water. The reason for this inconsistent result could be due to differences in the storage levels of L-AA in the tissues of the birds. It is well documented that the in ovo injection of L-AA into the amnion improves the antioxidant capacity of 42-day-old broilers [4]. However, serum L-AA levels have not been previously measured during the incubation period or the growing phase when L-AA was administrated by in ovo injection. The lack of an effect of L-AA supplementation on the serum L-AA concentrations of the embryos may also be related to the interactive effects of various physiological processes in the embryos, including their ability to synthesize L-AA, the turn-over and half-life of L-AA, and the continual loss of L-AA from the body pool. Concentrations of L-AA may also be higher in other tissues other than the blood, such as the spleen, liver, or eye. Future research is required to determine L-AA concentrations in the eye, liver, and spleen of broilers during the incubational and grow-out phases.

## 5. Conclusions

In conclusion, upon investigating the effects of high levels of L-AA injected into the amniotic sac on various hatch variables and serum L-AA levels, it was revealed that a high dose of L-AA (25 mg) did not have any negative effects on the hatch measurements that were examined. In addition, in consideration of the higher level of embryonic livability observed in the saline and 12 mg L-AA in ovo injection treatments in comparison to the non-injected controls, it can be concluded that the automated in ovo injection of high levels of L-AA may promote embryonic livability and not be detrimental to hatchling quality. Furthermore, serum L-AA levels did not change in response to the various dosages of L-AA administered by in ovo injection at 18 doi. Therefore, further research is needed to determine the retention levels of L-AA in the tissues of broilers during the grow-out phase when L-AA is administrated in ovo or in the diet.

## Figures and Tables

**Table 1 animals-12-01020-t001:** Effects of treatment non-injected; saline-injected (saline); saline containing 12 mg of L-ascorbic acid (L-AA 12), or 25 mg of L-ascorbic acid (L-AA 25) on percentage egg weight loss (PEWL) between 0 and 12, 12 and 18, and 0 and 18 days of incubation (doi), hatchability of injected live embryonated eggs (HI), and mean hatchling body weight (BW) at 21 doi ^1,2^.

	Treatment					
Items	Non-Injected	Saline	L-AA 12	L-AA 25	SEM	*p*-Value
0–12 PEWL (%)	4.61	4.57	4.63	4.66	0.071	0.644
12–18 PEWL (%)	4.18	3.83	4.16	4.16	0.294	0.603
0–18 PEWL (%)	8.61	8.23	8.59	8.62	0.295	0.502
HI (%)	93.1	91.3	93.6	91.3	2.75	0.765
Hatchling BW (g)	44.3	44.8	44.2	45.3	0.58	0.245

^1^ Injection volume = 100 μL. ^2^ Time of injection= 18 doi. *n* = approximately 30 eggs in each of 8 replicate groups in each treatment were used for means calculations.

**Table 2 animals-12-01020-t002:** Effects of treatment non-injected; saline-injected (saline); saline containing 12 mg of L-ascorbic acid (L-AA 12), or 25 mg of L-ascorbic acid (L-AA 25) on hatch residue analysis variables (late, pip, post-pip, and hatchling mortalities) at 21 days of incubation (doi) ^5,6^.

	Treatment					
Items	Non-Injected	Saline	L-AA 12	L-AA 25	SEM	*p*-Value
Late dead ^1^ (%)	2.80	4.81	4.85	5.94	1.466	0.218
Pipped dead ^2^ (%)	0.96	0.48	0.50	1.93	1.012	0.453
Pipped live ^3^ (%)	0.44	1.98	1.00	0.00	1.073	0.309
Dead chick ^4^ (%)	1.42 ^a^	0 ^b^	0 ^b^	0.47 ^a,b^	0.547	0.050

^a,b^ Treatment means for the same variable with no common superscript differ significantly (*p* < 0.05). ^1^ Mortality between 18 and 21 doi, prior to pip. ^2^ Mortality during the pipping process. ^3^ Mortality after the pipping process. ^4^ Mortality immediately after complete emergence of hatchlings from the shell. ^5^ injection volume = 100 μL. ^6^ Time of injection = 18 doi.

**Table 3 animals-12-01020-t003:** Effects of treatment non-injected; saline-injected (saline); saline containing 12 mg of L-ascorbic acid (L-AA 12), or 25 mg of L-ascorbic acid (L-AA 25) on serum L- ascorbic acid concentration at 18 and 19 doi, and male (L-AA-21-F doi) and female (L-AA-21-F doi) at 21 doi ^1,2^.

Treatment	Non-Injected	Saline	L-AA 12	L-AA 25	SEM	*p*-Value
L-AA-18 doi (μM)	7.18	6.67	7.41	7.03	0.568	0.630
L-AA-19 doi (μM)	11.43	12.03	11.78	11.17	2.248	0.982
L-AA-21-F doi (μM)	4.64	5.33	4.13	4.17	0.745	0.353
L-AA-21-M doi (μM)	8.62	8.63	10.82	6.86	0.325	0.307

^1^ injection volume = 100 μL. ^2^ Time of injection = 18 doi. *n* = Four eggs in each of eight replicate groups in each treatment were used for means calculations.

## Data Availability

None of the data were deposited in an official repository.

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
