# Peer review of "Effects of the In Ovo Injection of L-Ascorbic Acid on Broiler Hatching Performance†"

_animals, 2022, doi:10.3390/ani12081020_

Round 1

Reviewer 1 Report

All suggestions and comments were introduced in the PDF file. 

Reviewer 2 Report

Thank you for opportunity to review this manuscript. It is well written, results are clear. Methodology is well designed. I do not have major comments. I would  only suggest, that it would good to include in the observations, and thus the results, the survival of chicks and their condition during the first week after hatching.  As injections may had impact on their health during first days of postnatal growth.

Abstract

A total of 960 Ross 708 broilers were randomly divided into 4 treatment group – it is not clear. Do you mean adult broilers or eggs?

Blood samples were taken at 6 and 24 h after L-AA in ovo injection to determine serum – you took samples from embryos?

Introduction

I don’t have any comments. Introduction is informative, presenting the issues are very good as so, the literature. Aims and hypothesis is clear presented

Material and methods

1,440 broiler hatching eggs collected lub 960 eggs were involved into studies. Did you exclude them because of malformation etc.?

I have not other comments.

Results

No comments

Discussion

No comments

Round 2

Reviewer 1 Report

There are still some issues to be clarified. Moreover, the authors did not refer to some of my questions in Review 1. Regards

Author Response

Reviewer 1

1. The authors did not open the comment window. I did not mean to remove these words, but to replace them with a more appropriate one - "setting and hatching compartment" respectively.

I have additional questions to the authors (These questions were in the first review, but answers were omitted):

Answer: We sincerely apologize for missing and misunderstanding the comment. “Single stage” was inserted before “incubator” on line 122 to clarify single stage rather than multistage incubation. The wording of “setter and hatcher unit” is the appropriate terminology, as the whole incubator served as a setter unit between 0 and 18 days of incubation (doi) and as a hatcher unit between 18 and 21 doi. The single incubator was not compartmentalized into setter and hatcher areas. These are correct commonly used commercial terms. Please see Fatemi et al., 2021a,b (which are # 12 and 13 in the reference section) for the previously published use of this terminology.

Also, below are author’s responses for the following questions related to this section.

  1. There was only one incubation/hatching performed?

Answer: Yes, there was only one incubator that performed as a setter unit (0-18 doi) and as a hatcher unit (18-21 doi), and all hatch variables were evaluated at 21 doi.

  1. Why the authors decided to choose that late day of incubation? Whether did they base on other publications?

Answer: According to the literature the most successful time of injection is between 17.5 to 19 d of incubation. Also, the in ovo injection of L-AA at 17 doi has shown positive effects on live performance and antioxidant activity when it was injected in the amnion (Zhang et al., 2019).

  1. Did the authors think that this supplement was fully resorbed and utilised by chicks at the late stage of embryogenesis? It is difficult to evaluate without rearing of broilers.

Answer: Vitamin C is water-soluble vitamin and its absorption is rapid. Also, our laboratory has observed the long-term effects of the in ov injection of L-AA at different levels on live performance and antioxidant activity in grown broilers   

2. Do I mean by this that the entire incubation process took place in the same microclimate conditions? There were the same temperature and relative humidity from start to finish? If so, I am not convinced that it was correct. It is known that the temperature in the hatching compartment is lowered and the humidity is increased. Unfortunately, the cited work, Fatemi et. al (2020a), points to this. I am asking the authors to address this issue unequivocally. Moreover, I would suggest adding information about the specific microclimatic conditions (C and% RH) to the text.

Answer: Authors are in agreement that the temperature and RH are not the same for the setter and hatcher periods. The more appropriate reference now is used in this section. In the correct reference the incubational condition was reported as “For the setter phase of incubation, 37.5 °C dry bulb and 29. °C wet bulb temperatures were used, and for the hatcher phase, 36.9 °C dry bulb and 29.9 °C wet bulb temperatures were used”.

3. I agree, but the posthatch performance effect is just a guess about this research.

The effects of in ovo injection of 12 ml of L-AA have been shown to improve the live performance of broilers when hand injected (Zhang et al., 2018 and Zhang et al., 2019) and administered at 17 days of incubation. Also in Zhang et al (2019), the posthatch effect of L-AA on the antioxidant activity of broilers was increased in response to a high dosage of the in ovo L-AA. This effect was observed in grown broilers (42 d-old), not in younger birds, indicating the long-term effects of the in ovo injection of L-AA.

4. Line 281-284. The conclusion of the research results should only apply to what has been found. The authors continue to speculate in the last sentence. I will insist that this sentence should be removed as it does not relate directly to the obtained results.

The original sentence was” Nevertheless, it is suggested that dietary L-AA may be used to increase and maintain the retention levels of L-AA in the tissues of broilers during the grow-out phase.”

Answer: Thank you for the suggestion. The relevant correction was applied on lines 281-283. It is now presented as “Therefore, further research is needed to determine the retention levels of L-AA in the tissues of broilers during the grow-out phase when L-AA is administrated in ovo or in the diet.”   
